# Dynamic Pre-training: Towards Efficient and Scalable All-in-One Image Restoration

## Abstract

All-in-one image restoration tackles different types of degradations with a unified model instead of having task-specific, non-generic models for each degradation. The requirement to tackle multiple degradations using the same model can lead to high-complexity designs with fixed configuration that lack the adaptability to more efficient alternatives. We propose DyNet, a dynamic family of networks designed in an encoder-decoder style for all-in-one image restoration tasks. Our DyNet can seamlessly switch between its bulkier and lightweight variants, thereby offering flexibility for efficient model deployment with a single round of training. This seamless switching is enabled by our weights-sharing mechanism, forming the core of our architecture and facilitating the reuse of initialized module weights. Further, to establish robust weights initialization, we introduce a dynamic pre-training strategy that trains variants of the proposed DyNet concurrently, thereby achieving a 50% reduction in GPU hours. Our dynamic pre-training strategy eliminates the need for maintaining separate checkpoints for each variant, as all models share a common set of checkpoints, varying only in model depth. This efficient strategy significantly reduces storage overhead and enhances adaptability. To tackle the unavailability of large-scale dataset required in pre-training, we curate a high-quality, high-resolution image dataset named Million-IRD, having 2M image samples. We validate our DyNet for image denoising, deraining, and dehazing in all-in-one setting, achieving state-of-the-art results with 31.34% reduction in GFlops and a 56.75% reduction in parameters compared to baseline models. The source codes and trained models will be publicly released.

## 1 Introduction

The image restoration (IR) task seeks to improve low-quality input images. Despite several advancements in IR, diverse degradation types and severity levels present in images continue to pose a significant challenge. The majority of existing methods (Tu et al., 2022; Zamir et al., 2022; Wang et al., 2021; Zamir et al., 2021; 2020b; Chen et al., 2022) learn image priors implicitly, requiring separate network training for different degradation types, levels, and datasets. Further, these methods require a prior knowledge of image degradation for effective model selection during testing, thereby lacking generality to cater diverse degradations.

All-in-one restoration aims to restore images with an unknown degradation using a single *unified* model. Recent advancements, such as AirNet (Li et al., 2022) and PromptIR (Potlapalli et al., 2023), have addressed the all-in-one restoration challenge by employing contrastive learning and implicit visual prompting techniques, respectively. Specifically, the state-of-the-art PromptIR (Potlapalli et al., 2023) employs an implicit prompting to learn degradation-aware prompts on the decoder side, aiming to refine the decoder features. This approach, while intriguing, does not extend to the refinement of encoder features, which are left unprocessed. Despite the interesting idea of implicit prompting, it poses a considerable challenge for deployment due to its poor computational efficiency with 37M parameters and 243 GFlops to process a single 224×224 sized image. The practical application of such models is therefore challenging due to the high computational requirements, especially in resource-constraint hand-held devices. However, choosing lightweight models invariably leads to a trade-off between accuracy and efficiency (Zhang et al., 2020a; Li et al., 2023b). Most of these models aim to reduce parameter counts by strided convolutions and feature channel splitting (Kong et al., 2022; Li et al., 2023b; Kligvasser et al., 2018). Some strategies also operate within

 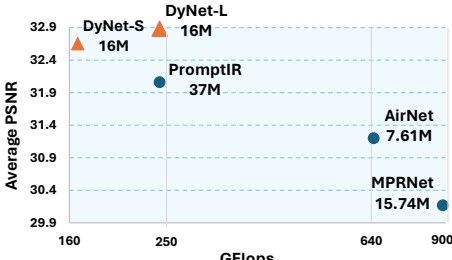

Figure 1: **Left:** At any given encoder-decoder level: (a) Transformer blocks in PromptIR, (b) and (c) Our DyNet-L and DyNet-S use the proposed weights sharing mechanism, initializing one transformer block and sharing its weights with subsequent blocks. **Right:** A plot of average PSNR in All-in-one IR setting vs GFlops and parameters (in millions). Our DyNet-S boosts performance by 0.43 dB, while reducing GFlops by 31.34% and parameters by 56.75% compared to PromptIR.

the frequency domain to reduce the computational demands linked to attention mechanisms (Zhou et al., 2023) or non-local operations (Guo et al., 2021) while few methods like (Cui & Knoll, 2023; Xie et al., 2021) partition the feature space for efficient processing, or split the attention across dimensions to improve the computational efficiency (Zhao et al., 2023; Cui & Knoll, 2023).

To optimize all-in-one IR efficiency without sacrificing performance, in this paper, we propose a novel weights-sharing mechanism. In this scheme, the weights of a network module are shared with its subsequent modules in a series. This approach substantially reduces the number of parameters, resulting in a more streamlined network architecture. We deploy the proposed weights-sharing mechanism in an encoder-decoder style architecture named Dynamic Network (DyNet). Apart from the computational efficiency, the proposed DyNet provides exceptional flexibility as by merely changing the module weights reuse frequency; one can easily adjust the network depth and correspondingly switch between its bulkier and lightweight variants. Model variants share the same checkpoints, differing only in depth, eliminating the need for separate checkpoints and saving disk space.

While DyNet offers great adaptability with orders of speedup, a common challenge with lightweight networks is the potential compromise on overall accuracy. To counteract this, we show that a large-scale pre-training strategy can effectively improve the performance of our lightweight models. By initializing the network with weights derived from the pre-training, the model benefits from a strong foundation, which can lead to enhanced performance even with a reduced parameter count. However, large-scale pre-training is computationally intensive and requires significant GPU hours. Therefore, we introduce an efficient and potent dynamic pre-training strategy capable of training both bulkier and lightweight network variants within a single pre-training session, thereby achieving significant savings of 50% in GPU hours. Complementing this, we have compiled a comprehensive, high-quality, high-resolution dataset termed Million-IRD, consisting of 2 million image samples. Our main contributions are as follows:

- We propose DyNet, a dynamic family of networks for all-in-one image restoration tasks. DyNet offers easy switching between its bulkier and light-weight variants. This flexibility is made possible through a weight-sharing strategy, which is the central idea of our network design, allowing for the efficient reuse of initialized module weights.

- We introduce a Dynamic Pre-training strategy, a new approach that allows the concurrent large-scale pre-training of both bulkier and light-weight network variants within a single session. This innovative strategy reduces GPU hours significantly by 50%, addressing the pressing challenge of resource-intensive model pre-training on a large-scale.

- We curate a comprehensive pre-training dataset comprising 2 million high-quality, high-resolution, meticulously filtered images. From this dataset, we extract 8 million non-overlapping high-resolution patches, each sized 512×512, utilized for the pre-training of variants of the proposed DyNet.

Through the synergy of these techniques, our proposed DyNet achieves an average gain of 0.68 dB for image de-noising, deraining, and dehazing within an all-in-one setting, with 31.34% reduction in GFlops and a 56.75% reduction in network parameters compared to the baseline; refer Fig. 1.

## 2 RELATED WORK

Image restoration (IR) seeks to restore images from their degraded low-quality versions, with the restoration process varying considerably across different tasks. Research in IR has predominantly concentrated on addressing single degradation challenges (Zamir et al., 2022; Ren et al., 2020; Dong et al., 2020a; Zamir et al., 2020a; Ren et al., 2021; Zhang et al., 2017b; Tsai et al., 2022; Nah et al., 2022; Zhang et al., 2020b). Although significant progress has been made in single degradation restoration techniques, research in multi-degradation restoration using a unified model is still relatively under-explored. Multi-degradation restoration, however, provides more practical advantages as the information about input degradation type is not required to select the task-specific model and it also enhances model storage efficiency over single-degradation restoration.

The critical challenge in multi-task IR is developing a single model capable of addressing various types of degradation and precisely restoring low-quality images. The IPT model (Chen et al., 2021), for instance, relies on prior knowledge of the corruption affecting the input image, utilizing a pre-trained transformer backbone equipped with distinct encoders and decoder heads tailored for multi-task restoration. However, in blind IR, we do not have such prior information about the degradation. In this direction, (Li et al., 2022) propose a unified network for multiple tasks such as denoising, deraining, and dehazing, employing an image encoder refined through contrastive learning. However, it requires a two-stage training process, where the outcome of contrastive learning relies on selecting positive and negative pairs and the availability of large data samples. Further, motivated by easy implementation and broad applicability of prompts, PromptIR (Potlapalli et al., 2023) introduces a concept where degradation-aware prompts are learned implicitly within an encoder-decoder architecture for all-in-one IR. PromptIR uses implicit prompting to refine decoder features but does not extend this refinement to the encoder features, leaving them unprocessed.

Although the above-discussed methods tackle multi-task IR, they suffer from low efficiency due to a large number of parameters and substantial GFlops consumption. Consequently, deploying these models in environments where efficiency is crucial factor becomes difficult due to their demanding computational needs. However, opting for lightweight models introduces an inevitable accuracy-efficiency trade-off (Zhang et al., 2020a). The primary cause is that existing methods (Kong et al., 2022; Li et al., 2023b; Kligvasser et al., 2018) focus on reducing parameter counts through techniques like strided convolutions and feature channel splitting. Further, few approaches work in frequency domain to reduce the computational burden associated with attention mechanisms (Zhou et al., 2023) or non-local operations (Guo et al., 2021). Furthermore, methods like (Cui & Knoll, 2023; Xie et al., 2021) partition the feature space for efficient processing, while (Zhao et al., 2023; Cui & Knoll, 2023) split the attention across dimensions to improve the computational efficiency. In contrast, our approach focuses on an efficient and scalable all-in-one image restoration model incorporating a weights-sharing mechanism. We demonstrate that strategic fundamental adjustments to the network architecture result in substantial performance improvements.

## 3 PROPOSED APPROACH

Existing SoTA all-in-one image restoration (IR) approaches offer a high computational footprint and do not provide the flexibility to change model complexity "on-the-go" during deployment. Towards more accurate, but lightweight and flexible architecture design for all-in-one IR, this work proposes a Dynamic Network Architecture (Sec. 3.1) and a Dynamic Pretraining Strategy (Sec. 3.2).

### 3.1 DYNAMIC NETWORK (DYNET) ARCHITECTURE

Our network architecture is based on a novel weight-sharing mechanism for IR models. This mechanism allows the network module's weights to be reused across subsequent modules in sequence, thereby significantly reducing the total number of parameters and leading to a more efficient network structure. We implement this weight-sharing approach within an encoder-decoder style architecture, which we term the Dynamic Network (DyNet). Within DyNet, at each encoder-decoder level, module weights are shared across a pre-specified number of subsequent modules. This not only enhances computational efficiency but also grants remarkable flexibility to the architecture. By simply altering the frequency of weight sharing, users can easily modify the network's depth, seamlessly transitioning between bulkier or lightweight variants of DyNet.

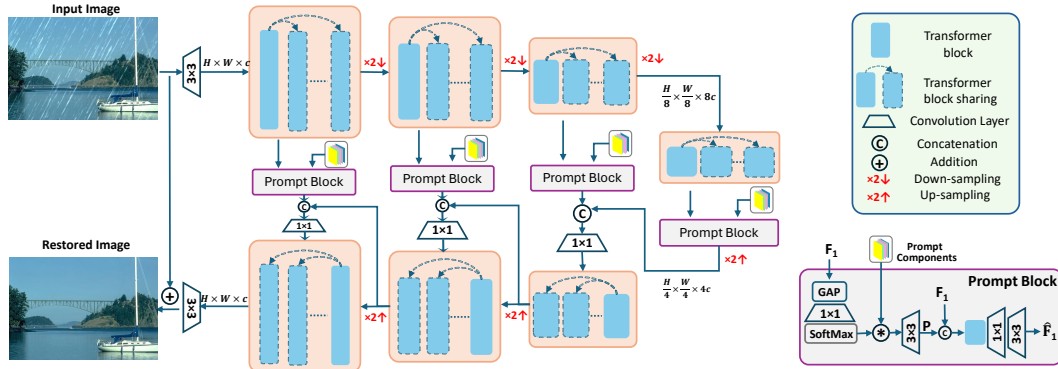

Figure 2: The proposed Dynamic Network (DyNet) pipeline for all-in-one image restoration. DyNet enhances a low-quality input image using a 4-level encoder-decoder architecture. A distinctive aspect of DyNet lies in its weight-sharing strategy. At each level, the initial transformer block's weights are shared with the subsequent blocks, significantly reducing the network parameters and enhancing its flexibility. This approach allows for easy adjustment of DyNet's complexity, switching between large and small variants by modifying the frequency of weight sharing across the encoder-decoder blocks. Moreover, we maintain the encoder-decoder feature consistency by implicitly learning degradation-aware prompts at skip connections rather than on the decoder side as in PromptIR.

### 3.1.1 OVERALL PIPELINE.

The proposed DyNet pipeline is shown in Fig. 2. The proposed DyNet begins by extracting low-level features $\mathbf{F_0} \in \mathbb{R}^{H \times W \times C}$ from a given degraded input image $\mathbf{I} \in \mathbb{R}^{H \times W \times 3}$ through a convolution operation. Subsequently, the feature embedding undergoes a 4-level hierarchical encoder-decoder, gradually transforming into deep latent features $\mathbf{F}_l \in \mathbb{R}^{\frac{H}{8} \times \frac{W}{8} \times 8C}$. We utilize the existing Restormer's (Zamir et al., 2022) Transformer block as a fundamental feature extraction module. For every level of the encoder and decoder, we initialize the weights for a first transformer block, which are subsequently reused across the subsequent blocks in a series up to the specified frequency at that level. For example, at a given encoder/decoder level, $w^1$ denotes weights of the first transformer block, we can define the output $\mathbf{F}_{out}$ of that encoder/decoder level, given input features $\mathbf{F}_{in}$ as,

$$\mathbf{F}_{out} = w^1(\mathbf{F}_{in}) \qquad for \ b = 1$$
$$\mathbf{F}_{out} = \circlearrowright_{b=2:f} w^b(\mathbf{F}_{out}) \qquad for \ b = 2:f$$

where, $\circlearrowright$ represents a self loop, iterate for $b$=2:$f$; $w^b = w^1 \ for \ b = 2, 3, \cdots, f$. Here, $f$ denotes the module weights reuse frequency for a given encoder or decoder level.

We gradually increase the reuse frequency of the transformer block from top level to the bottom level, thereby increasing the network depth. The multi-level encoder systematically reduces spatial resolution while increasing channel capacity, enabling the extraction of low-resolution latent features from a high-resolution input image. Subsequently, the multi-level decoder gradually restores the high-resolution output from these low-resolution latent features. To enhance the decoding process, we integrate prompt blocks from PromptIR (Potlapalli et al., 2023), which learn degradation-aware prompts implicitly through the prompt generation module (PGM) followed by the prompt-interaction module (PIM). Unlike the existing PromptIR, our approach incorporates degradation-aware implicit prompt blocks at skip connections. This implicit prompting via skip connections enables the transfer of refined encoder features to the decoder side, thereby enhancing the restoration process. This fundamental correction in the network design gives us significant improvement for all-in-one image restoration tasks, as detailed in the experimental section (Sec. 5).

### 3.1.2 ARCHITECTURAL DETAILS.

At each encoder-decoder level, by adjusting the module weights reuse frequency ($f$), we obtain DyNet's bulky and lightweight variants. Fig. 2 shows the core architecture of our DyNet. At each encoder-decoder level, we initialize weights ($w^1$) for the first transformer block and reuse them

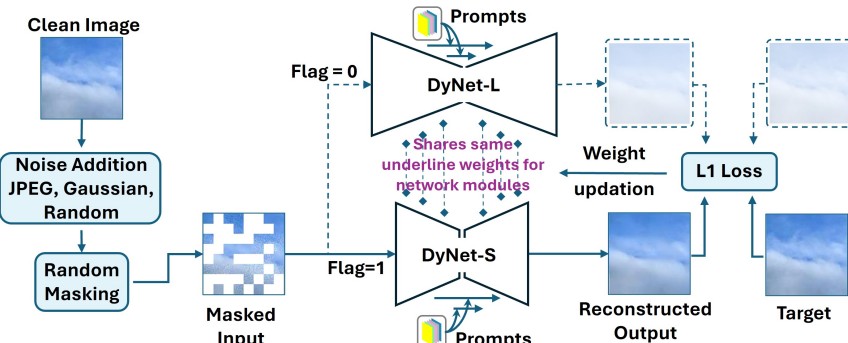

Figure 3: The proposed dynamic pre-training strategy is shown. Given a clean image, we create a degraded version by injecting noise (Gaussian or Random), JPEG artifacts and random masking within the same image. Two variants (small and large) of our DyNet are then trained concurrently to reconstruct the clean image from masked degraded inputs. Notably, the weights are shared between both variants since they are based on the same architecture but with an intra-network weight-sharing scheme with varying frequencies of block repetition. One of the two parallel branches is randomly activated in a single forward pass of the model (Flag being the binary indicator variable to show branch activation). The dotted lines show the network path is inactivated (Flag=0). An L1 loss is used to calculate pixel differences between reconstructed outputs and targets to update the shared weights of both branches. Both the inter and intra-model weight-sharing and random activation of branches lead to a significant reduction in GPU hours required for pre-training.

for the subsequent blocks. We vary the reuse frequency at each encoder-decoder level and obtain our bulkier and lightweight variants. DyNet comprises four encoder-decoder levels, with the larger variant (DyNet-L) employing reuse frequencies of $f = [4, 6, 6, 8]$ at encoder-decoder: level 1 to level 4, respectively. In contrast, the smaller variant (DyNet-S) applies weights reuse frequencies $f = [2, 3, 3, 4]$ for encoder-decoder level 1 to level 4, respectively. Further, we incorporate a prompt block at each skip connection, which enhances the encoder features being transferred to the decoder from the encoder. Overall, there are three prompt blocks: deployed one at each skip connection, each consisting of five prompt components each.

## 3.2 DYNAMIC PRE-TRAINING STRATEGY

In recent times, large-scale pre-training has become a key strategy to improve given network performance. Initializing the network with pre-trained weights provides a robust foundation, boosting performance even with fewer parameters. However, this strategy is resource-intensive, demanding considerable computational power and GPU hours. Therefore, we propose a dynamic pre-training strategy that enables simultaneous training of multiple network variants with shared module weights but varying depths. This allows for concurrent training of diverse models tailored to different computational needs and task complexities, all leveraging a shared architecture. Through the proposed dynamic pre-training strategy, we train both DyNet-L and DyNet-S variants of our DyNet concurrently in a single training session, achieving a 50% reduction in GPU hours.

DyNet-L and DyNet-S utilize the same underlying weights for transformer blocks, differing only in the reuse frequency of these blocks at each encoder-decoder level. Thus, during training iterations, we randomly alternate between DyNet-L and DyNet-S ensuring the optimization of the shared underlying weights as shown in Fig. 3. Moreover, we improve the generalization capability of our DyNet variants by adopting an input masking strategy akin to that proposed in masked auto-encoders (He et al., 2022). We randomly mask portions of an image and train the DyNet variants to reconstruct masked regions in a self-supervised fashion. The training dataset is described below.

## 4 MILLION-IRD: A DATASET FOR IMAGE RESTORATION

Large-scale pre-training for image restoration effectively demands large-scale, high-quality and high-resolution datasets. The current image restoration datasets: LSDIR (Li et al., 2023a),

Figure 4: **On the left:** Sample images from our Million-IRD dataset, which features a diverse collection of high-quality, high-resolution photographs. This includes a variety of textures, scenes from nature, sports activities, images taken during the day and at night, intricate textures, wildlife, shots captured from both close and distant perspectives, forest scenes, pictures of monuments, etc. **On the right:** Sample low-quality images filtered out during the data pre-processing phase (Sec. 4.1). These images were excluded due to being blurry, watermarked, predominantly featuring flat regions, representing e-commerce product photos, or being noisy or corrupted from artifacts.

NTIRE (Agustsson & Timofte, 2017), Flickr2K (Online), when combined, offer only a few thousand images. This is considerably inadequate compared to the extensively large-scale datasets LAION-5B (Schuhmann et al., 2022), ImageNet-21K (Ridnik et al., 2021) utilized in pre-training for other high-level tasks, such as visual recognition, object detection, and segmentation. The relatively small size of the existing training sets for image restoration restricts the performance capabilities of the underlying networks. More importantly, we are motivated by the scaling laws in high-level tasks that demonstrate the large-scale pre-taining can enable even lightweight, efficient model designs to reach the performance mark of much heavier models (Abnar et al., 2021). To address this gap, we introduce a new million-scale dataset named *Million-IRD* having ∼2M high-quality, high-resolution images, curated specifically for the pre-training of models for IR tasks. Fig. 4 shows few samples from our *Million-IRD* dataset. Our data collection and pre-processing pipeline are discussed below.

## 4.1 DATA COLLECTION AND PRE-PROCESSING

We combine the existing high-quality, high-resolution natural image datasets such as LSDIR (Li et al., 2023a), DIV2K (Agustsson & Timofte, 2017), Flickr2K (Online), and NTIRE (Ancuti et al., 2021). Collectively these datasets have 90K images with spatial size ranging between $< 1024^2, 4096^2 >$. Apart from these datasets, existing Laion-HR (Schuhmann et al., 2022) dataset has 170M unfiltered high-resolution images of average spatial size $1024^2$. However, these 170M samples are not directly suitable for model pre-training due to the predominant presence of low-quality images. Examples of such low-quality images are illustrated in Fig. 4 (on right side). Therefore, we focus on filtering out only high-quality images by discarding those of low quality. Given the impracticality of manual sorting of 170M images, we employ various image quality metrics NIQE (Mittal et al., 2012b), BRISQUE (Mittal et al., 2012a), and NIMA (Talebi & Milanfar, 2018). Only images surpassing certain predefined thresholds $T_{NIQE}$, $T_{BRISQUE}$, and $T_{NIMA}$ are selected; we empirically define thresholds for these metrics. With the mutual consensus of these metrics, we effectively filter out low-quality images having poor textural details, excessive flat areas, various artifacts like blur and noise, or unnatural content. By processing nearly $100M$ images in the Laion-HR dataset (Schuhmann et al., 2022), we sort out $2M$ high-quality images having spatial resolution of $1024^2$, or above. Examples of these high-quality filtered images are presented in Fig 4.

## 4.2 DATA POST-PROCESSING

Overall, our Million-IRD dataset has 2.09 million high-quality, high-resolution images. Each image in our dataset comes with metadata detailing its download link and the image resolution. A detailed breakdown of images and their respective sources are provided in the Appendix A. We extract high-resolution non-overlapping patches (of spatial size $512^2$) from each image, followed by the application of a flat region detector which eliminates any patch comprising more than 50% flat

area. This post-processing phase allows us to assemble a pool of $\sim$8 million diverse image patches, tailored for the pre-training phase of our DyNet variants.

# 5 EXPERIMENTS

We validate the proposed DyNet across three key image restoration tasks: dehazing, deraining, and denoising. In line with the existing PromptIR (Potlapalli et al., 2023), our experiments are conducted under two distinct setups: (a) All-in-One, wherein a single model is trained to handle all three degradation types, and (b) Single-task, where individual models are trained for each specific image restoration task.

## 5.1 IMPLEMENTATION DETAILS

**Dynamic Pre-training.** For robust weights initialization, we conduct a dynamic pre-training for two variants of our DyNet, i.e., DyNet-L and DyNet-S. Both variants share the same underline weights but differ in their transformer block reuse frequency at each encoder-decoder level. For the encoder-decoder levels 1 to 4, we set transformer block weights reuse frequencies of [4,6,6,8] for DyNet-L and [2,3,3,4] for DyNet-S. We use all $\sim$8M patches of size $512^2$ from our Million-IRD dataset for the dynamic pre-training. We randomly crop a $128^2$ region from each patch, employing a batch size of 32. Each of these cropped patches undergoes a random augmentation to 50% of its area using either JPEG compression, Gaussian noise, or random noise with varying levels. Furthermore, we mask 30% of the patch, leaving the remaining portion unaltered. In total, 80% of the input patch undergoes modification, either through degradation or masking. For weight optimization, we use the L1 loss and Adam optimizer with parameters $\beta_1 = 0.9$ and $\beta_2 = 0.999$, setting the learning rate to 1e-4 for 1 million iterations. Using the setup described above, we simultaneously pre-train both DyNet-L and DyNet-S. At any given iteration, we randomly switch between the two variants. Intriguingly, with each iteration, the shared underlying weights are optimized as shown in Fig. 3. As a result, by the end of this single pre-training session, we get DyNet-L and DyNet-S sharing the same trained underlying weights but differ in network depth, making them suitable for various challenges, including robustness and efficiency.

**Datasets.** For the all-in-one task implementations, we adopt the training protocol from PromptIR and concurrently fine-tune the pre-trained DyNet-S and DyNet-L in a single session. Following PromptIR, we prepare datasets for various restoration tasks. Specifically, for image denoising, we combine the BSD400 (Arbelaez et al., 2011) (400 training images) and WED (Ma et al., 2016) (4,744 images) datasets, adding Gaussian noise at levels $\sigma \in [15, 25, 50]$ to create noisy images. Testing is conducted on the BSD68 (Martin et al., 2001) and Urban100 (Huang et al., 2015) datasets. For deraining, we use Rain100L (Yang et al., 2020) with 200 training and 100 testing clean-rainy image pairs. Dehazing employs the SOTS (Li et al., 2018) dataset, featuring 72,135 training and 500 testing images. To build a unified model for all tasks, we merge the datasets and deploy a dynamic finetunning of the pre-trained DyNet for 120 epochs, then evaluate it across multiple tasks. The resulting flexible DyNet allows "on-the-go" adjustments to model depth, switching between DyNet-L and DyNet-S without needing separate model weights. Both versions share the same underlying weights, differing only in depth. For individual tasks, the pre-trained DyNet-L is fine-tuned for 120 epochs on the corresponding task-specific training set.

## 5.2 COMPARISONS ON MULTIPLE DEGRADATIONS UNDER ALL-IN-ONE SETTING

We benchmark our DyNet against a range of general IR methods and specific all-in-one solutions, as shown in Table 1. On average, across various tasks, DyNet-L and DyNet-S outperform the previously best PromptIR by $0.68$ dB and $0.43$ dB, respectively, and DyNet-S also cuts down 56.75% of parameters and 31.34% of GFlops. For image denoising, DyNet-L achieves a $1.11$ dB higher average PSNR compared to DL, and Fig. 5 showcasing its ability to deliver noise-free images with improved structural integrity. Notably, DyNet-L sets a new benchmark in image deraining and dehazing with improvements of 2.32 dB and 0.76 dB in PSNR, respectively. In addition, the visual comparisons in Fig. 6 demonstrate DyNet's superior capability to remove rain, producing cleaner images than PromptIR. For image dehazing, existing methods such as PromptIR are limited to removing neutral color (gray) haze. However, in real-world conditions, haze can appear in different colors, such as

Table 1: Comparison results in the All-in-one restoration setting. Our DyNet-L model outperforms PromptIR by 0.68 dB on average across tasks. Further, our DyNet-S model achieves a 0.43 dB average improvement over PromptIR, with reductions of 31.34% in parameters and 56.75% in GFlops.

| Comparative Methods | Dehazing on SOTS | Deraining on Rain100L | Denoising on BSD68 dataset | | | Average PSNR/SSIM |
|---|---|---|---|---|---|---|
| | | | $\sigma = 15$ | $\sigma = 25$ | $\sigma = 50$ | |
| BRDNet (Tian et al., 2020) | 23.23/0.895 | 27.42/0.895 | 32.26/0.898 | 29.76/0.836 | 26.34/0.836 | 27.80/0.843 |
| LPNet (Gao et al., 2019) | 20.84/0.828 | 24.88/0.784 | 26.47/0.778 | 24.77/0.748 | 21.26/0.552 | 23.64/0.738 |
| FDGAN (Dong et al., 2020b) | 24.71/0.924 | 29.89/0.933 | 30.25/0.910 | 28.81/0.868 | 26.43/0.776 | 28.02/0.883 |
| MPRNet (Zamir et al., 2021) | 25.28/0.954 | 33.57/0.954 | 33.54/0.927 | 30.89/0.880 | 27.56/0.779 | 30.17/0.899 |
| DL (Fan et al., 2019) | 26.92/0.391 | 32.62/0.931 | 33.05/0.914 | 30.41/0.861 | 26.90/0.740 | 29.98/0.875 |
| AirNet (Li et al., 2022) | 27.94/0.962 | 34.90/0.967 | 33.92/0.933 | 31.26/0.888 | 28.00/0.797 | 31.20/0.910 |
| PromptIR (Potlapalli et al., 2023) | 30.58/0.974 | 36.37/0.972 | 33.98/0.933 | 31.31/0.888 | 28.06/0.799 | 32.06/0.913 |
| **DyNet-S (Ours)** | 30.80/0.980 | 38.02/0.982 | 34.07/0.935 | 31.41/0.892 | 28.15/0.802 | 32.49/0.918 |
| **DyNet-L (Ours)** | 31.34/0.980 | 38.69/0.983 | 34.09/0.935 | 31.43/0.892 | 28.18/0.803 | 32.74/0.920 |

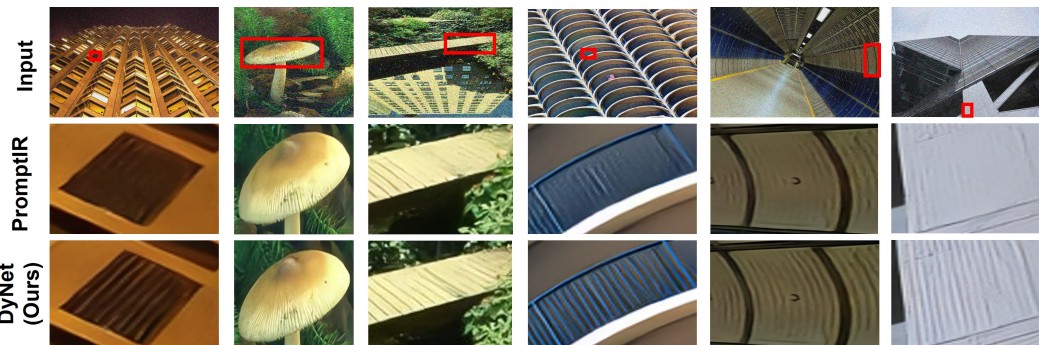

Figure 5: Comparative analysis of image denoising by all-in-one methods on the BSD68 and Urban100 dataset. DyNet reduces noise, produces a sharp and clear images compared to the PromptIR

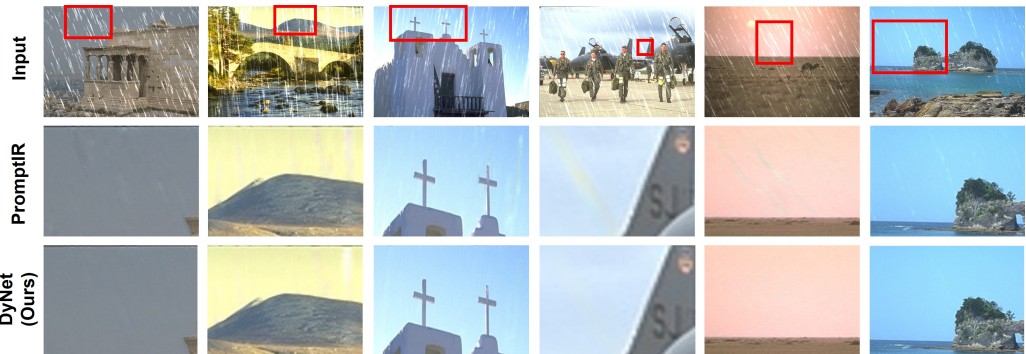

Figure 6: Comparative analysis within all-in-one methods for image de-raining on the Rain100L dataset. Our DyNet-L outperforms PromptIR by producing clearer, rain-free images.

light blue during the morning or yellow in smog-filled environments, etc. As illustrated in Fig. 7, PromptIR struggles to restore images with nongray haze due to its rigidity to haze-color types. Considering different types of haze (increasing the prompt length) during training could potentially address this issue. However, considering all types of haze color is not a practical solution. Therefore, to enhance our DyNet's ability to restore different haze-color images, we first restore the color balance using the existing Gray-World Assumption (GWA) algorithm followed by the dehazing through our network, which effectively recovers the haze-free image. We present a visual comparison between PromptIR and our DyNet on a variety of real-world hazy images in Fig. 7. For a fair assessment, we also include results from GWA+PromptIR (PromptIR+). The results clearly demonstrate that our DyNet outperforms PromptIR in reducing haze effects. The simple plug-in GWA module enables our approach to dehaze real-world hazy images without increasing model complexity.

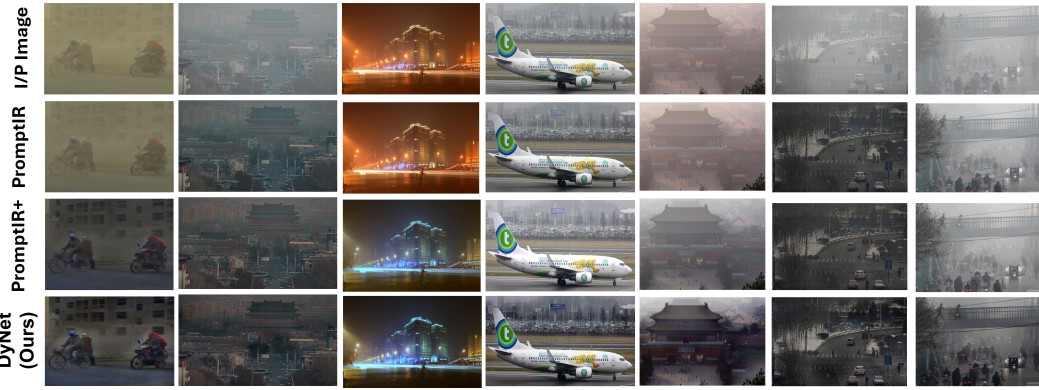

Figure 7: Comparative analysis of image dehazing by all-in-one methods on the real-world hazy images. Our approach reduces haze, producing more clear image compared to the PromptIR.

Table 2: Dehazing results in the single-task setting on the SOTS (Li et al., 2018) dataset. Our DyNet-L achieves a boost of 0.76 dB over PromptIR.

| MSCNN (Ren et al., 2020) | AODNet (Cai et al., 2016) | EPDN (Qu et al., 2019) | FDGAN (Dong et al., 2020b) | AirNet (Li et al., 2022) | Restormer (Zamir et al., 2022) | PromptIR (Potlapalli et al., 2023) | **DyNet-L** (Ours) |
|---|---|---|---|---|---|---|---|
| 22.06/0.908 | 20.29/0.877 | 22.57/0.863 | 23.15/0.921 | 23.18/0.900 | 30.87/0.969 | 31.31/0.973 | **32.07/0.982** |

Table 3: Deraining results in the single-task setting on Rain100L (Yang et al., 2020). Compared to the PromptIR, our method yields 1.81 dB PSNR improvement.

| DIDMDN (Zhang & Patel, 2018) | UMR (Yasarla & Patel, 2019) | SIRR (Wei et al., 2019) | MSPFN (Jiang et al., 2020) | LPNet (Gao et al., 2019) | AirNet (Li et al., 2022) | Restormer (Zamir et al., 2022) | PromptIR (Potlapalli et al., 2023) | **DyNet-L** (Ours) |
|---|---|---|---|---|---|---|---|---|
| 23.79/0.773 | 32.39/0.921 | 32.37/0.926 | 33.50/0.948 | 33.61/0.958 | 34.90/0.977 | 36.74/0.978 | 37.04/0.979 | **38.85/0.984** |

Table 4: Denoising comparisons in the single-task setting on BSD68 and Urban100 datasets. For difficult noise level of $\sigma = 50$ on Urban100, our DyNet-L obtains 0.75 dB gain compared to AirNet.

| Comparative Methods | Urban100( (Huang et al., 2015)) | | | BSD68( (Martin et al., 2001)) | | | Average PSNR/SSIM |
|---|---|---|---|---|---|---|---|
| | $\sigma = 15$ | $\sigma = 25$ | $\sigma = 50$ | $\sigma = 15$ | $\sigma = 25$ | $\sigma = 50$ | |
| CBM3D (Dabov et al., 2007) | 33.93/0.941 | 31.36/0.909 | 27.93/0.840 | 33.50/0.922 | 30.69/0.868 | 27.36/0.763 | 30.79/0.874 |
| DnCNN (Zhang et al., 2017a) | 32.98/0.931 | 30.81/0.902 | 27.59/0.833 | 33.89/0.930 | 31.23/0.883 | 27.92/0.789 | 30.74/0.878 |
| IRCNN (Zhang et al., 2017b) | 27.59/0.833 | 31.20/0.909 | 27.70/0.840 | 33.87/0.929 | 31.18/0.882 | 27.88/0.790 | 29.90/0.864 |
| FFDNet (Zhang et al., 2018) | 33.83/0.942 | 31.40/0.912 | 28.05/0.848 | 33.87/0.929 | 31.21/0.882 | 27.96/0.789 | 31.05/0.884 |
| BRDNet (Tian et al., 2020) | 34.42/0.946 | 31.99/0.919 | 28.56/0.858 | 34.10/0.929 | 31.43/0.885 | 28.16/0.794 | 31.44/0.889 |
| AirNet (Li et al., 2022) | 34.40/0.949 | 32.10/0.924 | 28.88/0.871 | 34.14/0.936 | 31.48/0.893 | 28.23/0.806 | 31.54/0.897 |
| Restormer (Zamir et al., 2022) | 34.67/0.969 | 32.41/0.927 | 29.31/0.878 | 34.29/0.937 | 31.64/0.895 | 28.41/0.810 | 31.79/0.903 |
| PromptIR (Potlapalli et al., 2023) | 34.77/0.952 | 32.49/0.929 | 29.39/0.881 | 34.34/0.938 | 31.71/**0.897** | **28.49/0.813** | 31.87/0.902 |
| **DyNet-L (Ours)** | **34.91/0.953** | **32.68/0.930** | **29.63/0.884** | **34.34/0.938** | **31.71**/0.896 | 28.47/0.812 | **31.96/0.902** |

## 5.3 COMPARISONS ON SINGLE DEGRADATION

We assess DyNet-L's efficacy in single-task settings, where distinct models are tailored to specific restoration tasks, demonstrating the effectiveness of content-adaptive prompting through prompt blocks. Our results, as shown in Table 2, indicate DyNet-L's superior performance with an improvement of 0.76 dB over PromptIR and 1.2 dB over Restormer in dehazing. This pattern is consistent across other tasks, including deraining and denoising. Notably, compared to PromptIR, DyNet-L achieves performance gains of 1.81 dB in deraining (refer to Table 3) and achieves a 0.24 dB improvement in denoising at a noise level of $\sigma = 50$ on the Urban100 dataset (Table 4).

## 5.4 ABLATION STUDIES

To study the impact of different components, we conduct various ablation experiments in the all-in-one setting, as summarized in Table 5. Table 5(a) illustrates that our DyNet-L and DyNet-S achieves

Table 5: Ablation experiments for different variants of the proposed DyNet.

| Comparative Methods | Dehazing on SOTS | Deraining on Rain100L | Denoising on BSD68 dataset (Martin et al., 2001) | | | Average | Par | GFlops |
|---|---|---|---|---|---|---|---|---|
| | | | $\sigma = 15$ | $\sigma = 25$ | $\sigma = 50$ | | | |
| PromptIR | 30.58/0.974 | 36.37/0.972 | 33.98/0.933 | 31.31/0.888 | 28.06/0.799 | 32.06/0.913 | 37M | 242.355 |
| **(a) Results without masked dynamic pre-training on Million-IRD** | | | | | | | | |
| **DyNet-S** | 30.10/0.976 | 37.10/0.975 | 33.94/0.931 | 31.28/0.880 | 28.03/0.789 | 32.09/0.910 | 16M | 166.38 |
| **DyNet-L** | 30.67/0.977 | 37.68/0.975 | 33.97/0.933 | 31.31/0.889 | 28.04/0.797 | 32.33/0.914 | 16M | 242.35 |
| **(b) Results with masked dynamic pre-training on Million-IRD** | | | | | | | | |
| **DyNet-S** | 30.80/0.980 | 38.02/0.982 | 34.07/0.935 | 31.41/0.892 | 28.15/0.802 | 32.49/0.918 | 16M | 166.38 |
| **DyNet-L** | **31.34/0.981** | **38.69/0.983** | **34.09/0.935** | **31.43/0.892** | **28.18/0.803** | **32.74/0.920** | 16M | 242.35 |

Table 6: Performance of the proposed DyNet-S on different combinations of degradation types.

| Degradation | | | Denoising on BSD68 dataset | | | Deraining on Rain100L | Dehazing on SOTS |
|---|---|---|---|---|---|---|---|
| Noise | Rain | Haze | $\sigma = 15$ | $\sigma = 25$ | $\sigma = 50$ | | |
| ✓ | ✗ | ✗ | 34.28/0.938 | 31.64/0.897 | 28.42/0.813 | - | - |
| ✗ | ✓ | ✗ | - | - | - | 38.69/0.983 | - |
| ✗ | ✗ | ✓ | - | - | - | - | 30.72/0.980 |
| ✓ | ✓ | ✗ | 34.18/0.936 | 31.52/0.893 | 28.31/0.808 | 38.54/0.983 | - |
| ✓ | ✗ | ✓ | 34.11/0.935 | 31.44/0.893 | 28.18/0.802 | - | 31.08/0.981 |
| ✗ | ✓ | ✓ | - | - | - | 38.12/0.982 | 30.92/0.980 |
| ✓ | ✓ | ✓ | 34.07/0.935 | 31.41/0.892 | 28.15/0.802 | 38.02/0.982 | 30.80/0.980 |

comparable performance to the baseline PromptIR with reductions of 56.75% in parameters and 31.34% in GFlops, without the proposed dynamic pre-training on Million-IRD dataset. This shows that the proposed fundamental correction of placing prompt block at skip connections is effective compared to on the decoder side as given in PromptIR. Furthermore, pre-training on our Million-IRD dataset brings a significant improvement, resulting in an average PSNR increase of 0.41 dB for DyNet-L compared to the DyNet-L without pre-training as shown in Table 5(b). A similar effect is observed for our DyNet-S variant. Moreover, our proposed dynamic training strategy reduces GPU hours by 50% when training variants of the DyNet. Overall, each of our contributions (*i.e.*, implicit prompting via skip connections, dynamic network architecture, Million-IRD dataset, and dynamic pre-training strategy) synergistically enhances the performance of the proposed approach, while significantly reducing GFlops and learning parameters compared to the baseline PromptIR.

**Training model with different combinations of degradation.** In Table 6, we compare the performance of our DyNet-S in all-in-one setting on aggregated datasets, assessing how varying combinations of degradation types affect it's effectiveness. All the models in this ablation study are trained for 80 epochs. Here in Table 6, we assess how varying combinations of degradation types (tasks) influence the performance of our DyNet. Notably, the DyNet trained for a combination of the derain and de-noise tasks exhibits better performance for image deraining than the DyNet trained on the combination of derain and dehaze. This shows that some degradations are more relevant for the model to benefit from when trained in a joint manner.

## 6  CONCLUSION

This paper introduces a novel weight-sharing mechanism within the Dynamic Network (DyNet) for efficient all-in-one image restoration tasks, significantly improving computational efficiency with a performance boost. By adjusting module weight-reuse frequency, DyNet allows for seamless alternation between bulkier and lightweight models. The proposed dynamic pre-training strategy simultaneously trains bulkier and lightweight models in a single training session. Thus saving 50% GPU hours compared to traditional training strategy. The ablation study shows that the accuracy of both bulky and lightweight models significantly boosts with the proposed dynamic large-scale pre-training on our Million-IRD dataset. Overall, our DyNet significantly improves the all-in-one image restoration performance with a 31.34% reduction in GFlops and a 56.75% reduction in parameters compared to the baseline models.

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

## A    APPENDIX (SUPPLEMENTARY MATERIAL)

Here, we have discussed the details about the transformer block used in our DyNet, additional sample images from our Million-IRD dataset, additional visual results comparison between PromptIR (Potlapalli et al., 2023) and the proposed DyNet in all-in-one setting.

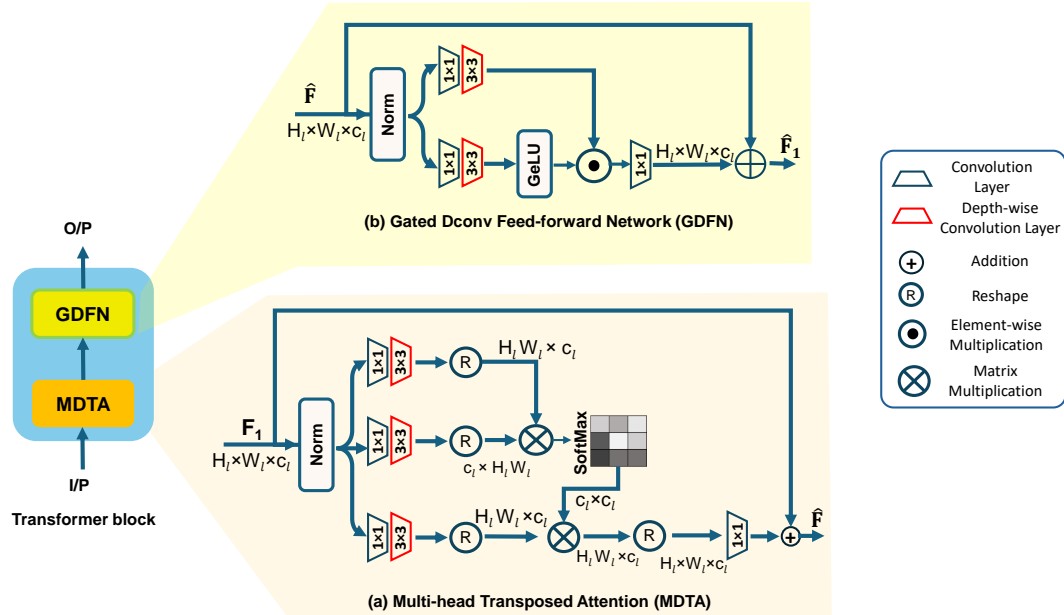

Figure 8: Overview of the Transformer block used in our DyNet network. The Transformer block is composed of two sub-modules, the Multi Dconv head transposed attention module (MDTA) and the Gated Dconv feed-forward network (GDFN).

## B  TRANSFORMER BLOCK DETAILS

As described in Section 3.1 of the main manuscript, here, we have discussed the transformer block utilized within our DyNet network architecture, detailing its sub-modules multi deconv head transposed attention (MDTA) and gated deconv feed-forward network (GDFN). Initially, input features, represented as $R \in H_l \times W_l \times C_l$, are processed through the MDTA module. Within this, Layer normalization is used to normalize the input features. This is followed by the application of $1 \times 1$ convolutions and then $3 \times 3$ depth-wise convolutions, which serve to transform the features into Query (Q), Key (K), and Value (V) tensors. A key feature of the MDTA module is its focus on calculating attention across the channel dimensions, instead of the spatial dimensions, which significantly reduces computational demands. For channel-wise attention, the Q and K tensors are reshaped from $H_l \times W_l \times C_l$ to $H_l W_l \times C_l$ and $C_l \times H_l W_l$ dimensions, respectively. This reshaping facilitates the computation of the dot product and leads to a transposed attention map of $C_l \times C_l$ dimensions. This process incorporates bias-free convolutions and executes attention computation simultaneously across multiple heads. Following the MDTA Module, the features undergo further processing in the GDFN module. Within this module, the input features are initially expanded by a factor of $\gamma$ through the use of $1 \times 1$ convolutions. Subsequently, these expanded features are processed with $3 \times 3$ convolutions. This procedure is executed along two parallel pathways. The output from one of these pathways is subjected to activation through the GeLU non-linear function. The activated feature map is then merged with the output from the alternative pathway via an element-wise multiplication.

## C  VISUAL RESULTS COMPARISON

We show additional visual results comparison between PromptIR (Potlapalli et al., 2023) and our DyNet under all-in-one setting.

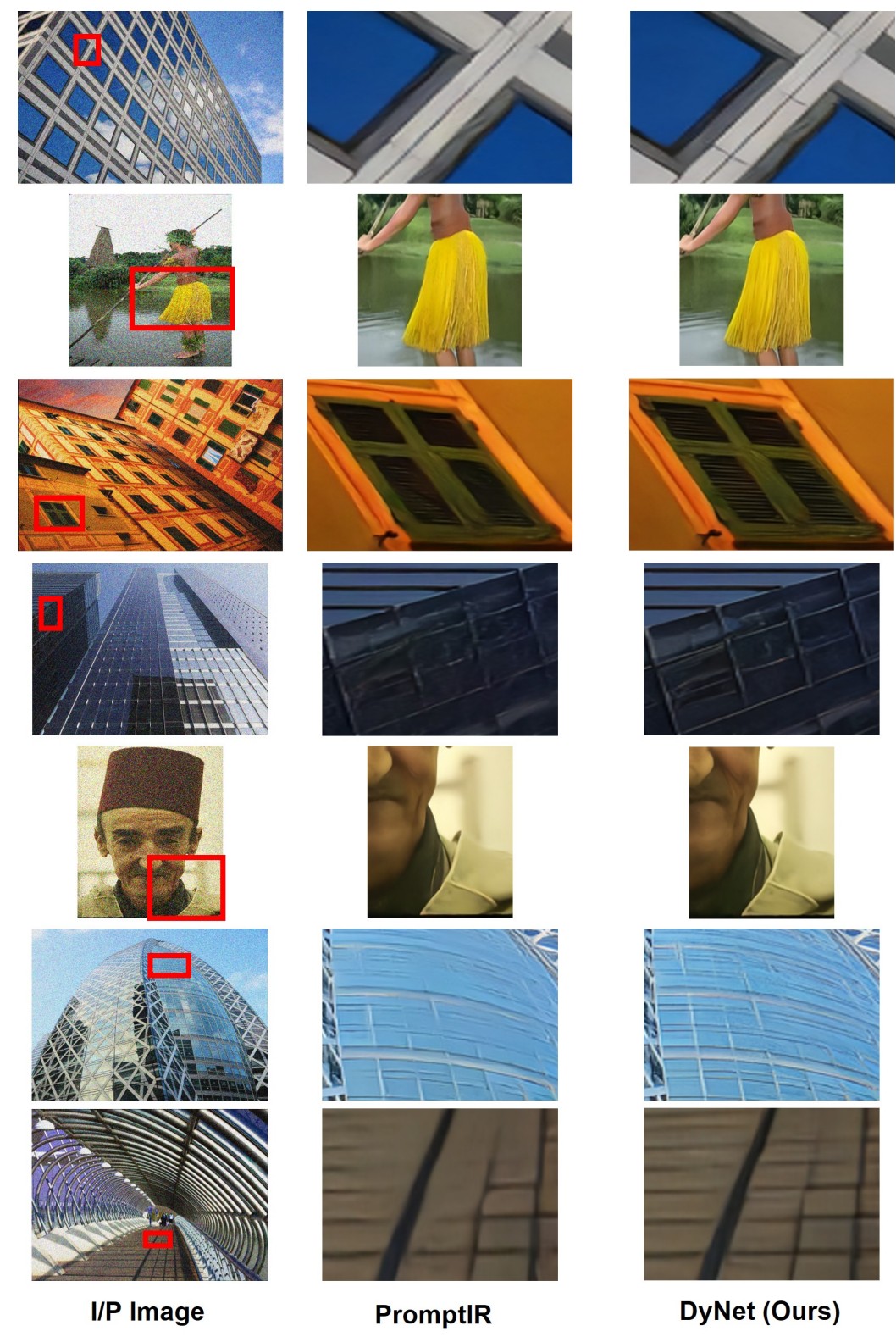

Figure 9: Comparative analysis of image denoising by all-in-one methods on the BSD68 dataset (Martin et al., 2001) and Urban100 (Huang et al., 2015). Our DyNet reduces noise, producing more sharper and clearer image compared to the PromptIR (Potlapalli et al., 2023).

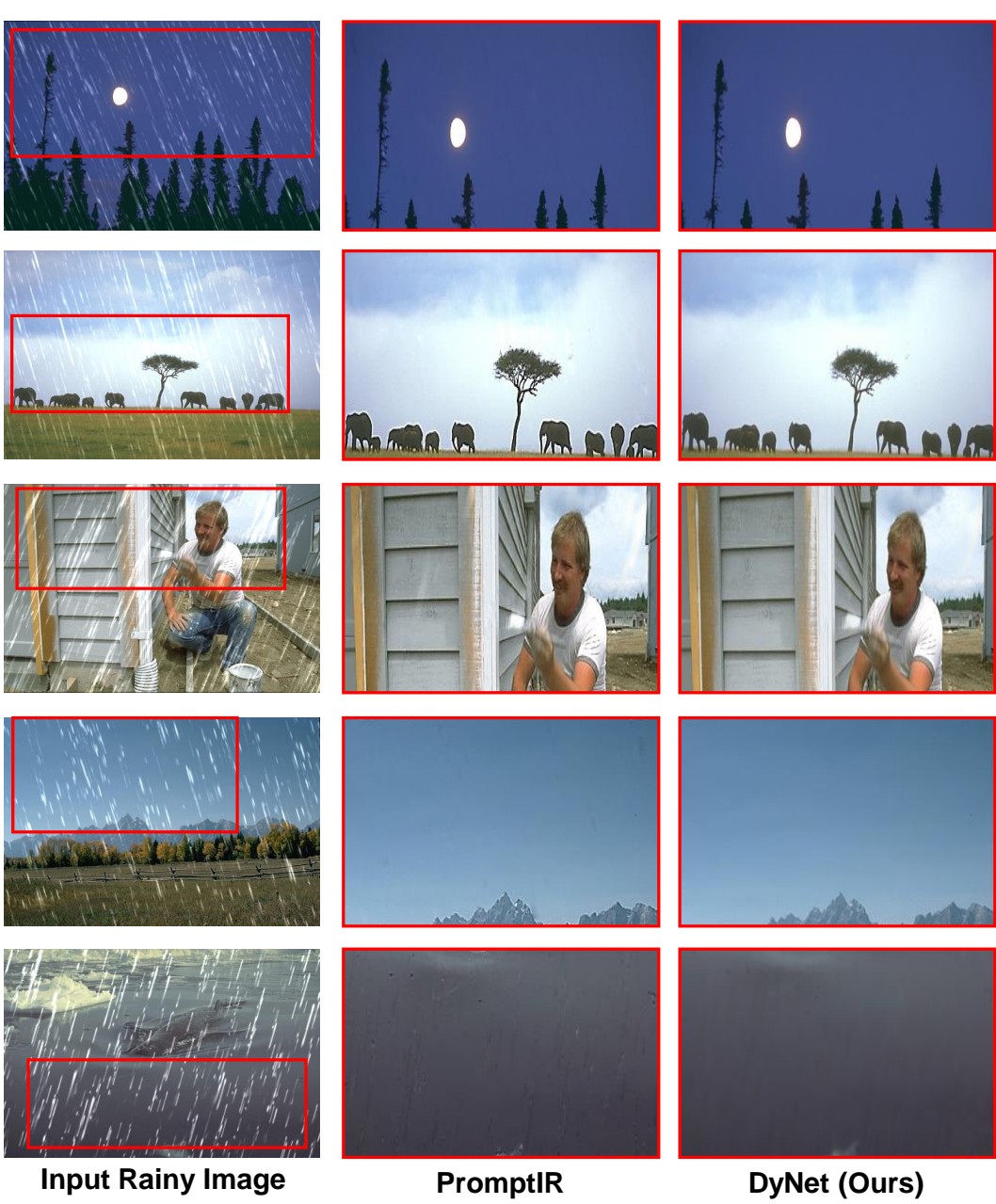

**Input Rainy Image**          **PromptIR**          **DyNet (Ours)**

Figure 10: Comparative analysis of image deraining by all-in-one methods on the Rain100L dataset (Fan et al., 2019). Our DyNet-L successfully eliminates rain streaks, producing clear, rain-free images as compared to the recent PromptIR Potlapalli et al. (2023).

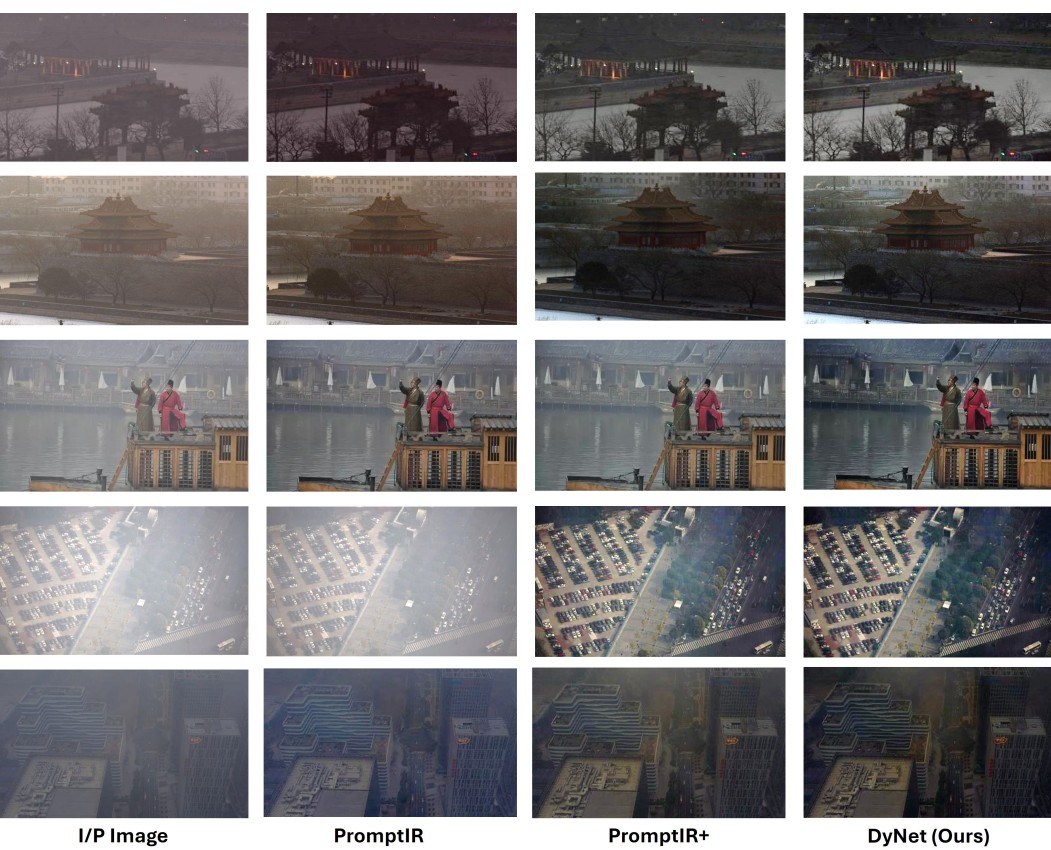

I/P Image PromptIR PromptIR+ DyNet (Ours)

Figure 11: Comparative analysis of image dehazing by all-in-one methods on the SOTS dataset (Li et al., 2018). Our approach reduces haze, producing more clear image compared to the PromptIR (Potlapalli et al., 2023).

Table 7: Overview of the number of images curated from each database for our Million-IRD dataset.

| Dataset | NTIRE | DIV2K | Flikr2K | LSDIR | Laion-HR |
|---------|-------|-------|---------|-------|----------|
| (Ancuti et al., 2021) | (Agustsson & Timofte, 2017) | (Online) | (Li et al., 2023a) | (Schuhmann et al., 2022) | |
| #Images | 56 | 2,000 | 2,000 | 84,991 | 2M |

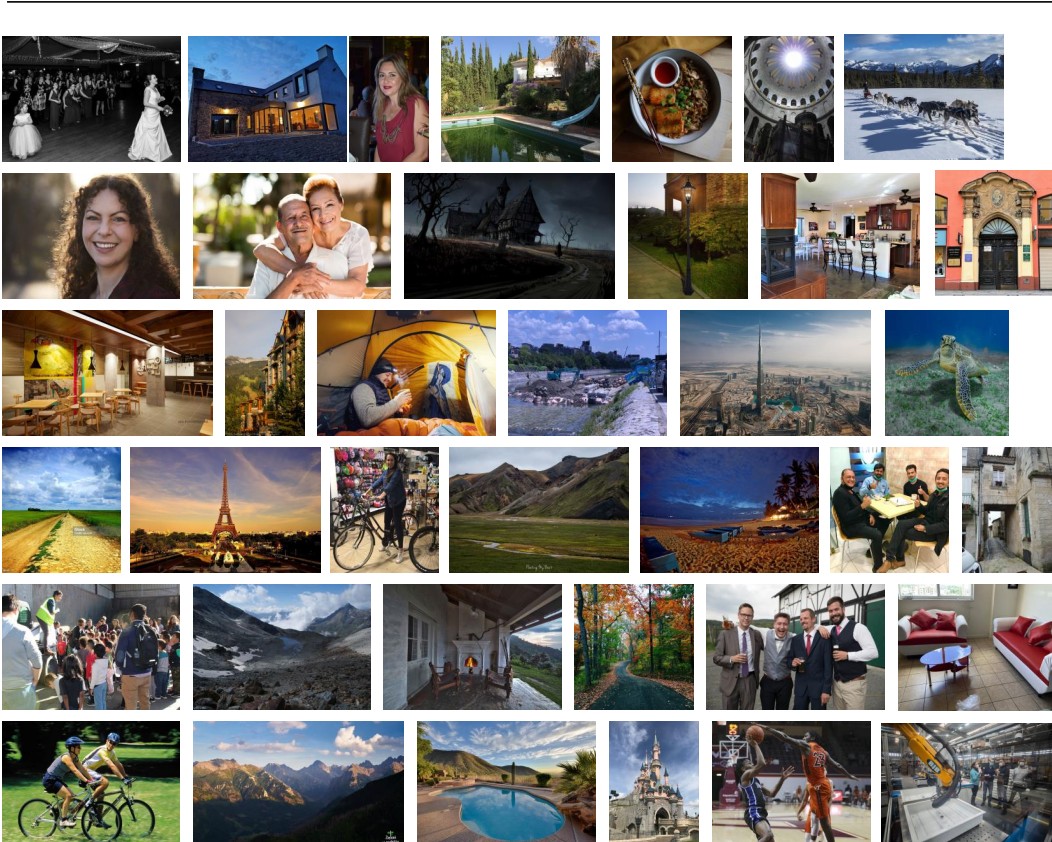

Figure 12: Sample images from our Million-IRD dataset, which features a diverse collection of high-quality, high-resolution photographs. This includes a variety of textures, scenes from nature, sports activities, images taken during the day and at night, intricate textures, wildlife, shots captured from both close and distant perspectives, forest scenes, pictures of monuments, etc.

## D BREAKDOWN OF IMAGES IN OUR MILLION-IRD DATASET

We combine the existing high-quality, high-resolution natural image datasets such as LSDIR (Li et al., 2023a), DIV2K (Agustsson & Timofte, 2017), Flickr2K (Online), and NTIRE (Ancuti et al., 2021). Collectively these datasets have 90K images having spatial size ranging between $< 1024^2, 4096^2 >$. Their breakdown is given in Table 7. We also show the additional sample images from our Million-IRD dataset in Fig. 12 and Fig. 13.

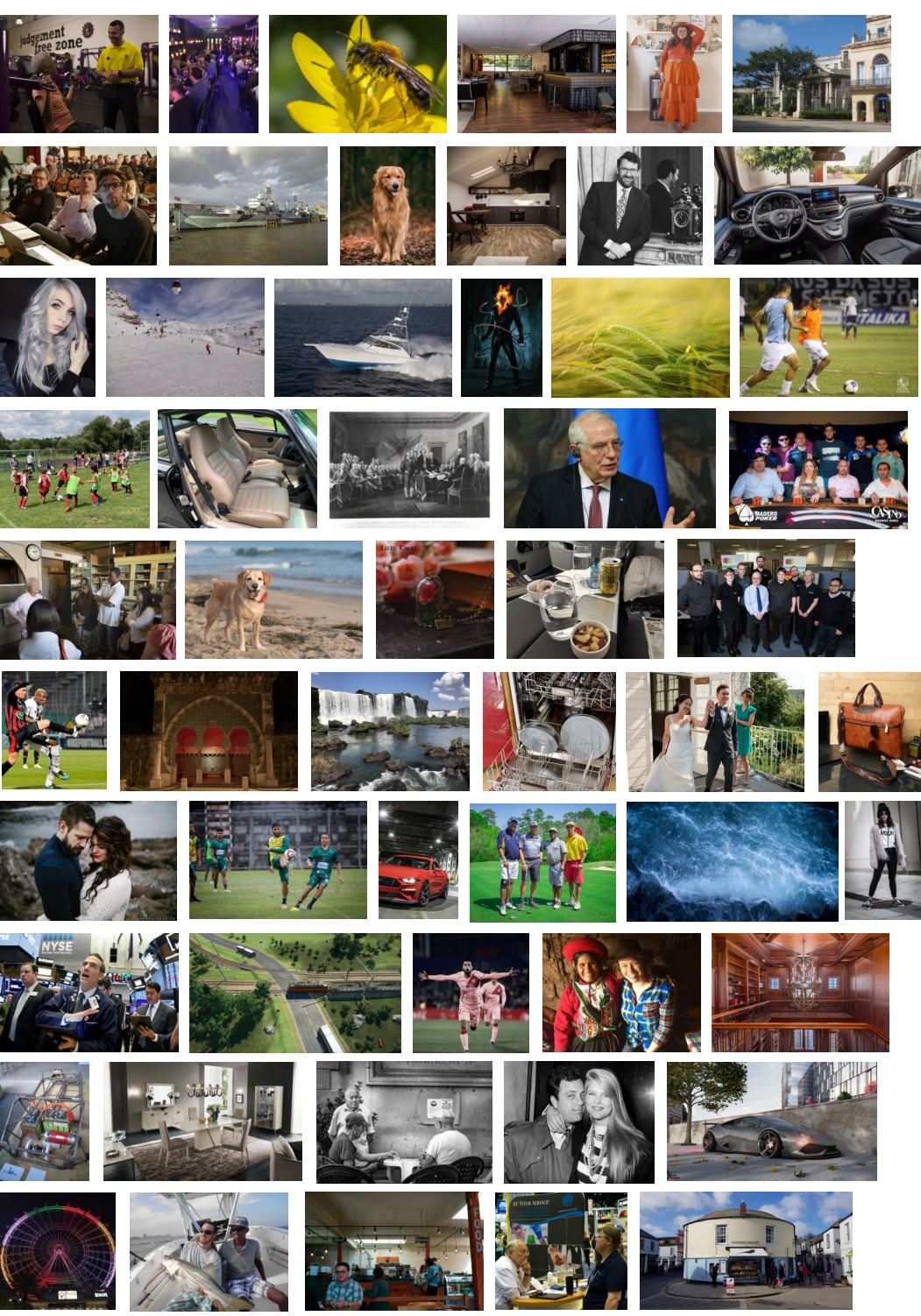

Figure 13: Sample images from our Million-IRD dataset, which features a diverse collection of high-quality, high-resolution photographs. This includes a variety of textures, scenes from nature, sports activities, images taken during the day and at night, intricate textures, wildlife, shots captured from both close and distant perspectives, forest scenes, pictures of monuments, etc.

