# OpenReview forum: "Dynamic Pre-training: Towards Efficient and Scalable All-in-One Image Restoration"
_ICLR.cc/2025/Conference — ICLR 2025 Conference Withdrawn Submission_

### Official Review · Reviewer_boup · 2024-11-01

**Soundness:** 3
**Presentation:** 3
**Contribution:** 2
**Rating:** 5
**Confidence:** 4

**Summary:**

The paper presents several notable contributions to all-in-one image restoration. The proposed DyNet model features a weight-sharing mechanism, significantly reducing parameters and computational complexity—by 56.75% and 31.34% in GFlops, respectively—compared to previous methods. The innovative dynamic pre-training strategy allows simultaneous training of large and small model variants, saving about 50% of GPU hours. Additionally, the Million-IRD dataset, consisting of high-quality, high-resolution images, enhances the pre-training process and provides a strong foundation for performance improvement. Overall, DyNet achieves state-of-the-art results across tasks such as denoising, deraining, and dehazing, showing significant performance gains compared to baseline models. These features make DyNet suitable for deployment in resource-constrained environments.

**Strengths:**

1. The proposed dynamic pre-training strategy is a novel approach that simultaneously trains multiple model variants, significantly reducing GPU hours while maintaining performance. This approach introduces flexibility and efficiency not seen in prior work.
2. The paper presents the contributions with clear explanations of the technical details, particularly the weight-sharing and dynamic pre-training components. The architectural diagrams and comprehensive experimental analysis aid in understanding the model's advantages.
3. The design of DyNet with its weight-sharing mechanism ensures both computational efficiency and reduced parameter count. The integration of a large-scale, high-resolution dataset (Million-IRD) further demonstrates a high-quality, data-driven approach to improving model performance.

**Weaknesses:**

1. While DyNet demonstrates impressive results on common degradation types, the evaluation could benefit from including more diverse or challenging degradation scenarios to further establish its generalizability.
2. The ablation studies provide insights into the contributions of individual components; however, adding more analysis on the sensitivity of weight-sharing frequencies and dataset variations could strengthen the understanding of model robustness.

**Questions:**

1.  Can the authors provide further explanation regarding the choice to introduce a dynamic pre-training strategy? Although the strategy appears to reduce GPU hours, it would be beneficial to understand whether this approach also impacts model robustness or generalization across tasks. Could conventional pre-training approaches suffice in meeting efficiency goals?

---

### Official Review · Reviewer_hE7A · 2024-11-01

**Soundness:** 3
**Presentation:** 3
**Contribution:** 2
**Rating:** 5
**Confidence:** 5

**Summary:**

This paper presents DyNet, an efficient image restoration network addressing the All-in-One restoration setting on three degradations (haze, rain and noise).

The network is based on the popular restoration architectures Restormer/PromptIR, however proposes a weight sharing mechanism which can reduce the numbers of parameters by ~50% and GFLOPs by ~30% compared to PromptIR. DyNet contains a single Transformer block instance in each level of the UNet architecture, while the depth of the network is controlled by re-using the same block instance several times.

Besides, they propose a pre-training strategy in conjunction with a large-scale image dataset containing around 2M samples. To reduce the overall pre-training time of their framework, the authors train different model scales simultaneously, leveraging their proposed parameter sharing while varying the reuse frequency, which increases the network depth without adding additional parameters.

**Strengths:**

1. The idea is straightforward yet effective, enabling simultaneous training of two independent model scales that share the same parameters.

2. Compared to previous All-in-One methods, DyNet achieves a new state-of-the-art with a more compact model size and a reduced computational footprint.

3. The paper is clearly written and the dataset contribution might be beneficial for the community, as its size is significant.

**Weaknesses:**

1. The technical novelty of DyNet is limited, as it primarily mirrors the PromptIR architecture, with the main difference being the addition of weight sharing.

2. The ablation studies presented are not very convincing. Apart from Table 5, the experiments do little to illustrate or provide intuition into the proposed weight-sharing mechanism, which is the paper’s primary contribution (see questions).

3. The proposed pre-training approach requires further clarification. Currently, it’s unclear why training on "cheap" synthetic degradations, like noise and JPEG compression artifacts, would be beneficial for the All-in-One restoration task, which targets degradations such as haze, rain, and noise.

4. Although the authors mention an interest in scaling laws (line 295), they present results for only two model scales without addressing the amount of data necessary for effective pre-training.

**Questions:**

1. What are the intuitions behind using specific degradations in the pre-training process? Why not rely solely on high-ratio image masking?

2. How does the proposed pre-training approach differ from earlier methods in low-level vision, such as DegAE [ref1] or CoLoRA [ref2]?

3. Where are the limitations of the weight-sharing mechanism? Is it feasible to apply the 16M parameter model at a smaller or larger scale? Could we also start with a smaller parameter model and then scale it up?

4. Is the weight sharing mechanism applicable to other restoration architectures, such as SwinIR [ref3] or CAT [ref4]?

5. Does the pre-training help with model generalization for real degradations or more complex scenarios such as the 5 degradations All-in-One setting (haze, rain, noise, motion blur and low illumination)?


[ref1] Liu et al., DegAE: A New Pretraining Paradigm for Low-Level Vision, CVPR 2023

[ref2] Park et al., CoLoRA: Contribution-based Low-Rank Adaptation with Pre-training Model for Real Image Restoration, ECCV 2024

[ref3] Liang et al., SwinIR: Image Restoration Using Swin Transformer, ICCVW 2021

[ref4] Chen et al., Cross Aggregation Transformer for Image Restoration, NEURIPS 2022

---

### Official Review · Reviewer_n5Zd · 2024-11-01

**Soundness:** 3
**Presentation:** 3
**Contribution:** 2
**Rating:** 5
**Confidence:** 5

**Summary:**

This paper propose DyNet, a dynamic network family for image restoration tasks, allowing easy switching between larger and lightweight variants through a weight-sharing strategy that efficiently reuses initialized module weights. DyNet also introduces a Dynamic Pre-training strategy, enabling large-scale pre-training of both network types concurrently within a single session, reducing GPU hours. This work also contain a dataset of 2 million high-quality, high-resolution images.

**Strengths:**

1. The proposed method can indeed get two models with shared weights and different sizes in one training session. Although there are no relevant experiments, it may also be applicable to various network structures.
2. The proposed pre-training method and the new data set help the network achieve better performance than the previous models.
3. A new large scale dataset is good for the community.

**Weaknesses:**

I think the article has weaknesses in the following aspects.


1. The model claims to save 50% training time, but I feel that the application scenarios are rare. The model is actually training a large model and then obtain a small variant at the same time. If we only need one model (whether large or small), we only need one training process. Only when we need multiple sizes of models at the same time can we use this approach to save computation. Then, even if we need multiple models, especially if we need a small model, we can take some other tiny structure (such as CNN) and do fewer training iters, there is no need to get two models at the same time. Besides that, since it does not affect the inference time, increasing the training time and the number of parameters is acceptable in the current application. Can the author give some scenarios where it is necessary to get two similar variants at the same time?


2. Many key questions about the method are not provided with experiments. The current experiments focus only on improving performance.

     (1) The model can get two models at the same time, can it be three or four? When there are more models, can we not increase the number of parameters? There are no relevant experiments in this paper.


     (2) The frequency of training switching (maybe the -S model only need less training frequency). No related exploration is provided. Further, the method also can be compared with some other alternatives to improve the practicability of the method, such as training a large and a small model, using half of the training time in the paper, respectively, observing the difference between them.


     (3) The pre-training experiment in table 5 actually couples the effect of the pre-training method and the dataset. I'm not sure if pre-training with more data or using masked pre-training resulted in an improvement. It would have been better to provide a model pre-training on the new dataset but did not use masked training.

     (4) The DyNet method mentioned in this paper has weak relationship with the pre-training method, it seems that they are two methods that can be used separately (in Table5), and there is no related discussion in this paper.


3. I think that the finetune dataset have some problems. There is a large gap in the number of images between different tasks in finetune (during training 7w for dehaze and 200 for derain), which may affect the accuracy of some conclusions (e.g., in table 6). (This may not be the author's problem, just follow previous works, but I think this thing needs thinking)

**Questions:**

Pleace see Weaknesses.

---

### Official Review · Reviewer_bWtJ · 2024-11-03

**Soundness:** 3
**Presentation:** 1
**Contribution:** 1
**Rating:** 1
**Confidence:** 4

**Summary:**

This paper presents a transformer-based model for all-in-one image restoration. The proposed method incorporates weight-sharing and weight-initialization (or pretraining) techniques with a new high-quality dataset. Experimental results show that the proposed methods outperform previous methods in terms of PSNR and FLOPs efficiency.

**Strengths:**

The training efficiency of the proposed method is reasonable.

The proposed methods outperform the previous all-in-one restoration models regarding PSNR and FLOPs efficiency.

**Weaknesses:**

The ablation studies are limited to showing the effectiveness of each component of the proposed techniques in diverse measures.
For instance, Table 5 presents the effectiveness of masked dynamic pre-training on Million-IRD. Still, it is not clear whether the performance improvement is due to the pre-training technique or the training dataset.

The novelty of the proposed method is marginal, with a limited literature review.
Weight-sharing and pre-training are well-known techniques in neural network training for image classification and restoration, but this paper presents limited previous works and comparisons in this area.

Experiments do not support the contributions of this paper well, and the contributions are not significant.
The first contribution includes "DyNet offers easy switching between its bulkier and light-weight variants." However, the experiments show only two types of models, DyNet-S and DyNet-L.
The second contribution is about training efficiency through pre-taining, but experiments do not analyze the proposed method in training efficiency.
The last contribution concerns a new dataset, but the experiments do not address why the dataset is important in all-in-one restoration models.

**Questions:**

How do the low-quality images, rejected at the pre-processing stage, affect the restoration performance?

---

### Note · Authors · 2024-11-20

I have read and agree with the venue's withdrawal policy on behalf of myself and my co-authors.